# Omega-3 Fatty Acids during Pregnancy in Indigenous Australian Women of the Gomeroi Gaaynggal Cohort

**DOI:** 10.3390/nu15081943

**Published:** 2023-04-18

**Authors:** Natalie L. Gray, Isobel Stoodley, Lisa G. Wood, Clare E. Collins, Leanne J. Brown, Kym M. Rae, Kirsty G. Pringle, Tracy L. Schumacher

**Affiliations:** 1School of Health Sciences, University of Newcastle, Callaghan, NSW 2308, Australiaclare.collins@newcastle.edu.au (C.E.C.); 2School of Biomedical Sciences and Pharmacy, University of Newcastle, Callaghan, NSW 2308, Australia; isobel.stoodley@uon.edu.au (I.S.); lisa.wood@newcastle.edu.au (L.G.W.); kirsty.pringle@newcastle.edu.au (K.G.P.); 3Immune Health Program, Hunter Medical Research Institute, New Lambton, NSW 2305, Australia; 4Food and Nutrition Research Program, Hunter Medical Research Institute, New Lambton, NSW 2305, Australia; leanne.brown@newcastle.edu.au; 5Department of Rural Health, University of Newcastle, Tamworth, NSW 2340, Australia; 6Mater Medical Research Institute, South Brisbane, QLD 4101, Australia; kym.rae@uq.edu.au; 7Faculty of Medicine, University of Queensland, Herston, Brisbane, QLD 4072, Australia; 8Mothers and Babies Research Program, Hunter Medical Research Institute, New Lambton, NSW 2308, Australia

**Keywords:** Omega-3, pregnancy, Indigenous, nutrition, preterm birth, hypertension, nutrient supplement, *n*-3 LC-PUFAs, eicosapentaenoic acid, docosahexaenoic acid

## Abstract

Higher dietary intakes of Omega-3 long-chain polyunsaturated fatty acids (*n*-3 LC-PUFAs) have been linked to lower rates of preterm birth and preeclampsia. The aim of this analysis was to describe dietary intake and fractions of red blood cell (RBC) membrane LC-PUFAs during pregnancy in a cohort of Indigenous Australian women. Maternal dietary intake was assessed using two validated dietary assessment tools and quantified using the AUSNUT (Australian Food and Nutrient) 2011–2013 database. Analysis from a 3-month food frequency questionnaire indicated that 83% of this cohort met national *n*-3 LC-PUFA recommendations, with 59% meeting alpha-linolenic acid (ALA) recommendations. No nutritional supplements used by the women contained *n*-3 LC-PUFAs. Over 90% of women had no detectable level of ALA in their RBC membranes, and the median Omega-3 Index was 5.5%. This analysis appears to illustrate a decline in concentrations of eicosapentaenoic acid (EPA) and docosahexaenoic acid (DHA) across gestation in women who had preterm birth. However, there was no visible trend in LC-PUFA fractions in women who experienced hypertension during pregnancy. Further research is needed to better understand the link between dietary intake of *n*-3 LC-PUFA-rich foods and the role of fatty acids in preterm birth and preeclampsia.

## 1. Introduction

Long-chain polyunsaturated fatty acids (LC-PUFAs) are a group of fatty acids (FAs) that are necessary for the normal growth and development of both mothers and babies during pregnancy. Much is known about the role of Omega-3 (*n*-3) LC-PUFAs in fetal development, as they support the development of the fetal nervous system and retina [1,2]. The *n*-3 LC-PUFAs also support the mother’s overall health through the physically challenging time of pregnancy. They are known to play an important structural role within cell membranes, strengthen cognitive function, and lower cardiovascular disease risk factors via their anti-inflammatory properties [3,4,5]. There is also a growing body of evidence linking higher levels of *n*-3 LC-PUFA consumption with improved outcomes during pregnancy, including lower rates of preterm birth, preeclampsia, and perinatal depression [6,7]. The *n*-3 LC-PUFAs are unable to be synthesised by the body efficiently [8], and for this reason, it is vital that women consume adequate amounts from their diet or via supplementation throughout pregnancy.

The *n*-3 LC-PUFA, alpha-linolenic acid (ALA), acts as the precursor to endogenous formation of the longer chain variants, eicosapentaenoic acid (EPA), docosapentaenoic acid (DPA), and docosahexaenoic acid (DHA) [8]. The biological conversion of ALA to EPA, DPA, and DHA uses the same desaturase enzymes that convert the Omega-6 (*n*-6) FA, linoleic acid (LA), to its long-chain variants such as arachidonic acid (AA) [8]. As a result, endogenous synthesis of EPA, DPA, and DHA is considered inefficient, highly variable, and dependent on the overall dietary FA profile [8]. The ratio of LA to ALA is often higher in Western diets, resulting in greater amounts of *n*-6 LC-PUFAs produced in comparison to the *n*-3 LC-PUFA derivatives [8]. Therefore, weekly consumption of pre-formed EPA, DPA, and DHA from the diet is recommended to ensure that intake is sufficient. The FAs EPA, DPA, and DHA are found primarily in marine food sources such as salmon, tuna, bream, trout, eel, and sardines [9]. ALA, as the precursor to the longer *n*-3 LC-PUFAs, is mostly found in plant food sources such as vegetable oils, margarine, and linseed oils [9].

Australian Nutrient Reference Values (NRVs) recommend nutrient intake targets across the lifespan and are based on existing scientific evidence [9]. Due to a lack of available data, there are no Recommended Dietary Intake (RDI) or Estimated Average Requirement (EAR) levels set for *n*-3 LC-PUFA intakes in Australia. In this circumstance, an Adequate Intake (AI) is assigned, which represents estimated nutrient adequacy based on the intake of healthy populations. For pregnant women in Australia aged 19–50 years, the AI target is 10 g/day for LA, 1.0 g/day for ALA, and 115 mg/day for total EPA, DPA, and DHA consumption [9]. For EPA, DPA, and DHA, this is equivalent to approximately two servings of fatty fish per week [10]. Studies report that pregnant women commonly do not meet these requirements due to concerns about contaminants, including mercury levels in fish, a neurotoxin that can have harmful effects on the fetus [11,12]. Women are often unaware of the health benefits of including these *n*-3 LC-PUFA-rich foods in their diet during pregnancy, and conflicting messages often lead to women avoiding fish altogether to prevent any potential harm to the fetus [11,12,13]. Additionally, women may not be consuming enough fish during pregnancy due to access, cost, pregnancy-related food aversions, general dislike, and/or family preferences [11].

There is very little data available on the intake of *n*-3 LC-PUFAs during pregnancy for Australian women and for specific population groups such as Aboriginal and Torres Strait Islander women (hereafter respectfully termed Indigenous). As a result of colonisation and the subsequent social and health inequities in Australia, Indigenous women are more likely to experience poor antenatal health outcomes when compared to non-Indigenous women [14]. This is highlighted by preterm birth rates in Australia, with 14% of Aboriginal and Torres Strait Islander babies born preterm, compared to 8.7% of births in the overall population [15,16]. The Australian Aboriginal and Torres Strait Islander Health Survey results of 2012–2013 reported that Indigenous Australian women of childbearing age were meeting the AI for *n*-3 LC-PUFA intake during pregnancy with an average daily ALA intake of 1.3 g, and 204 mg of total EPA, DPA, and DHA [17]. These figures, however, do not specifically report intakes during pregnancy separately.

The primary aim of this paper was to describe the dietary and supplemental intake of LC-PUFAs in a cohort of Indigenous Australian women during their pregnancy. This analysis uses data from the Gomeroi Gaaynggal cohort, a longitudinal study examining the health of Indigenous Australian women and their children during pregnancy and beyond. While previous studies of the Gomeroi Gaaynggal cohort have analysed overall maternal diet quality and intakes of several key nutrients during pregnancy [18,19], LC-PUFAs have not been reported. The secondary aim is to describe the fractions of LC-PUFAs from red blood cell (RBC) membranes over the course of pregnancy from a sample of women in the cohort and map the results with key pregnancy outcomes of hypertension during pregnancy (HDP) and preterm birth.

## 2. Materials and Methods

### 2.1. Study Design

The Gomeroi Gaaynggal study was a prospective longitudinal study of Indigenous and non-Indigenous women carrying Indigenous Australian babies throughout their pregnancy and into early childhood [20]. The study aimed to achieve meaningful and positive health outcomes for Indigenous mothers and their babies by examining a wide range of markers, including nutrition and chronic disease risk factors. The Gomeroi Gaaynggal cohort was established in close partnership with the local Indigenous communities involved [21]. Prior to recruitment commencing, an extensive 2-year community consultation process was undertaken to ensure that the objectives of the study were planned with communities to identify their interests [21]. A study plan was formalised as a result of these conversations. Recruitment of participants occurred between 2009–2019 and was facilitated by Indigenous members of the research team within antenatal clinics. In 2013, the Gomeroi Gaaynggal Advisory Committee was established to ensure a validated, Indigenous-led governance structure across all projects within the study. The Advisory Committee was consulted for initial approval to proceed with the analysis outlined in this paper in May 2021. Final approval was gained from the committee prior to the submission of the paper for publishing, with further input and consultation scheduled throughout all stages of the analysis. Additional details of the cohort study protocol have been previously published [20].

### 2.2. Ethics

Ethics approval for the Gomeroi Gaaynggal study was received from the following committees: Hunter New England Human Research Ethics Committee (HNEHREC No. 08/05/21/4.01); the University of Newcastle Human Research Ethics Committee (UONHREC No. H-2009-0177); and the Aboriginal Health and Medical Research Council Human Research Ethics Committee, NSW (AHMRC HREC No. 654/08).

### 2.3. Setting

Recruitment of the cohort took place in two towns in NSW, Australia. Town 1 covers an area of 9900 m^2^ and is located 400 km northwest of the nearest major city and 200 km inland from the east coast of Australia. It has a population of approximately 60,000 people, with a median weekly household income of $AUD 1180 [22]. Approximately 10% of residents identify as Indigenous, compared with the NSW average of 3%, and the Indigenous median weekly household income is $AUD 1106 [22]. Town 2 is located 550 km northwest of the nearest major city and 500 km from the east coast. It has approximately 2150 people, with 43% of those identifying as Indigenous [23]. The median weekly household income of the town is $AUD 1039, compared with $AUD 789 for its Indigenous residents [23].

### 2.4. Participants

Pregnant women who identified as Indigenous Australian, as well as non-Indigenous women carrying Indigenous babies, were eligible to enrol in the study at any stage of their pregnancy. Written informed consent was obtained from all study participants following consultation with Indigenous members of the research team to ensure each mother had a thorough understanding of the study.

### 2.5. Demographics and Other Health Factors

An online survey was administered to obtain additional demographic data from participants during study visits, with Indigenous members of the research team available for assistance if required. Questions were predominantly multiple-choice, with space to provide additional detail or explanation where necessary. Information was collected about age, Indigenous identity, gravidity and parity, income, education, employment, and any medical history related to obstetrics and reproductive health. Diabetes status, gestational age at delivery, and birth weight of the baby were obtained from hospital records or collected directly from participants where hospital records were not available. Smoking status was also established and considered positive if the participant reported smoking at any time during their pregnancy.

Participants self-reported their pre-pregnancy weight, and a member of the research team measured their current height and weight. The pre-pregnancy body mass index (BMI) of participants was then calculated [weight (kg)/height (m)^2^]. These results should be interpreted with caution as BMI measures have not been validated to accurately estimate overweight and obesity in Indigenous Australian populations [24]. Individual sub-categories within pre-pregnancy BMI and diabetes status were grouped and presented accordingly to protect the anonymity of participants.

### 2.6. Assessment of Pregnancy Outcomes

Participants were categorised as having HDP as reported on their antenatal hospital records. This included a diagnosis of gestational hypertension, preeclampsia, or eclampsia, with the three conditions grouped to protect the anonymity of participants. Where hospital records were unavailable, participants were defined as having preeclampsia based on blood pressure measurements at least 4 h apart post 20 weeks of gestation and before the onset of labour, in which systolic pressure was ≥140 mmHg and/or diastolic pressure ≥ 90 mmHg with proteinuria (urinary protein ≥ 300 mg/24 h or spot urine protein:creatinine ratio ≥ 30 mg/mmol creatinine or urine dipstick protein ≥ ++) [25]. Preterm birth was categorised as the birth of a live infant earlier than 37 weeks of gestation. This was determined by the gestational age reported on hospital antenatal records or calculated based on gestational age determined from ultrasound records and date of birth where hospital records were unavailable. Participants could be assigned to both the HDP and preterm birth outcome groups if they experienced both pathologies during their pregnancy.

### 2.7. Dietary and Supplemental Intake of Omega-3 Fatty Acids

Two validated dietary assessment tools were used for the nutritional analyses in this study, with all data collected between 2014–2019 [26,27]. Dietary assessments were not performed in the cohort prior to this time. Nutrient intakes from both dietary data sets have been quantified using the AUSNUT (Australian Food and Nutrient) 2011–2013 database, which was considered the most comprehensive source of nutrient information in Australia [28]. AI values were used to determine participants’ nutritional adequacy of *n*-3 FA intake during pregnancy, with intake from each tool reported separately.

A 24-h food recall was collected from participants during the earlier stages of their pregnancy. Prior to 2018, this data was obtained from participants via a structured interview with a qualified dietitian using the validated triple pass method [26]. The triple pass method ensures a comprehensive account of all foods consumed in the previous 24 h by asking about the participants’ intake at three different stages in the interview. This method provides a short-term, detailed view of intake, including data on individual food and beverage items and their quantities [26]. A qualified dietitian then entered this data as food records into the Australian version of the Automated Self-Administered 24-Hour (ASA24) Dietary Assessment Tool [29]. From 2018, dietary intake data has been self-recorded by the participant via the National Cancer Institute’s ASA24 (Australia), a validated multiple-pass method. Both a trained Indigenous research assistant and a qualified dietitian were with the participant to assist with any questions about the survey. Total energy (kJ), gram weight and nutrient content of individual food and supplement items were quantified. As supplements are often used during pregnancy to support increased nutritional requirements, the AUSNUT 2011–2013 Dietary Supplement Nutrient Database was used to categorise and account for supplementation in this analysis [9,28,30].

In addition, the Australian Eating Survey Food Frequency Questionnaire (AES FFQ) was used during the third trimester to assess dietary intake across the pregnancy. The AES FFQ is a self-administered tool that captures estimated dietary intake over the previous six months via 120 semi-quantitative questions [27]. The surveyed food list is thorough to allow estimation and ranking of usual macronutrient and micronutrient intakes, with *n*-3 LC-PUFA consumption measured from food items only [27]. A response for each food or food type is a frequency with options ranging from ‘never’ to ‘four or more times per day’. The AES FFQ has been shown to provide a valid and reliable estimate of dietary intakes of Australian adults (median age: females 41.3 years, males 44.9 years) over the previous six months [27], with validity demonstrated for *n*-3 LC-PUFA dietary intakes compared to red blood cell membrane FAs in both children and adults [31]. Further information about this survey has been previously published [27].

### 2.8. Red Blood Cell Sample Collection and Analysis

Blood samples were collected from participants during each trimester of their pregnancy by an Indigenous research assistant trained in phlebotomy. Random (non-fasting) samples were collected in EDTA tubes and stored on ice until centrifugation at 3000× *g* at 4 °C for 10 min. RBCs were separated and stored at −70 °C before analysis.

A sub-sample of participants from the cohort was chosen based on their preterm birth and HDP status. These women were then case-matched to those with an uncomplicated pregnancy based on their stage of the trimester, followed by age and pre-pregnancy BMI where possible.

#### 2.8.1. Erythrocyte Membrane Fatty Acid Preparation

Using the method described by Tomoda et al. [32], the erythrocytes were lysed, and their membranes were solubilised and purified. A total of 500 µL of erythrocytes were vortexed with 12 mL of hypotonic tris buffer (10 mM tris hydroxymethyamino methane/5 mM ascorbate buffer, pH 7.4). After standing on ice for five minutes, 12 mL of 0.25 M glucose solution was added and vortexed again. Following another five minutes on ice, the sample was centrifuged at 10,000 RPM at 4 °C for ten minutes. After discarding the supernatant, the procedure was repeated twice more (resuspending the pellet by vortexing) with the same quantities of tris and glucose solutions above, then centrifuged at 12,000 RPM at 4 °C for 10 min and then 15,000 RPM at 4 °C for 20 min. The pellet was then resuspended in 250 µL each of the tris and glucose solutions. The sample was stored at −80 °C prior to methylation.

#### 2.8.2. Fatty Acid Determination

Total erythrocyte membrane fatty acids were determined via the method established by Lepage and Roy [33]. Two mL of a methanol/toluene mixture (4:1 *v*/*v*) containing C21:0 (0.02 g/L) as an internal standard and BHT (0.12 g/L) was added to 500 µL of erythrocyte membrane suspension. A total of 200 µL of acetyl chloride was added dropwise to methylate the fatty acids. The samples were heated to 100 °C for one hour. After cooling, 5 mL of 6% potassium carbonate solution was added to stop the reaction. To facilitate the separation of the layers, the sample was centrifuged at 3000 RPM at 4 °C for 10 min. The upper toluene layer was used for gas chromatography analysis of the fatty acid methyl esters, using a 30 m × 0.25 m (DB-225) fused carbon-silica column coated with cyanopropylphenyl (J & W Scientific, Folsom, CA, USA). Both injector and detector port temperatures were set at 250 °C. The oven temperature was 170 °C for two minutes, increased 10 °C/min to 190 °C, held for one min (24), then increased 3 °C/min up to 220 °C and maintained to give a total run time of 30 min. A split ratio of 10:1 and an injection volume of 5 mL were used. The chromatograph was equipped with a flame ionisation detector, autosampler, and autodetector. Sample fatty acid methyl ester peaks were identified by comparing their retention times with those of a standard mixture of fatty acid methyl esters and quantified using a Hewlett Packard 6890 Series Gas Chromatograph with Chemstations Version A.04.02.

#### 2.8.3. Fatty Acid Calculations

Saturated, monounsaturated, polyunsaturated, *n*-3 LC-PUFA, and *n*-6 LC-PUFA are reported as a percentage of total FAs. Omega-3 Index was calculated as [(erythrocyte membrane EPA (mg) + erythrocyte membrane DHA (mg))/total erythrocyte FAs (mg)] × 100 [34]. The Omega-6:Omega-3 ratio was determined by dividing total Omega-6 FAs (%) by total Omega-3 FAs (%).

### 2.9. Statistical Analysis

A descriptive cross-sectional analysis of all dietary (24-h recall and FFQ) and maternal blood sample data obtained from the cohort was undertaken. Only data from participants carrying a single fetus and with either a dietary record or blood sample were used for the analysis. All data obtained were included, regardless of any diagnosed or suspected medical conditions. Where multiple 24-h recall entries were collected during a single pregnancy, the earliest record obtained was included. FFQ data were excluded if the overall energy intake was deemed implausible (<4500 kJ/day or >20,000 kJ/day) or if it was collected outside the period of 27 weeks of gestation to birth.

All data were tested for normality using Shapiro–Wilks deemed non-normal and presented as the median and interquartile range (IQR) or as a number and percentage where appropriate. Separate descriptive statistics were performed on the 24-h recall data to account for *n*-3 LC-PUFA supplementation.

Scatterplots were used to visually depict the fractions of key LC-PUFAs recorded in the RBC membranes (Total EPA + DPA + DHA, LA, ALA, EPA, DPA, and DHA). No power calculations are presented, as this is an exploratory analysis based on available data. Pregnancy outcomes were overlaid on this data, with a line of best fit to highlight any trends associated with either *n*-3 LC-PUFA intake and/or maternal blood levels of *n*-3 LC-PUFAs. Additionally, figures were used to visually represent the fractions of total EPA + DPA + DHA, LA, and ALA in the RBC membranes relative to the nutritional adequacy of LC-PUFA intake compared with AI values. Participants were categorised as having met the AI level if they met the AI in either or both the 24-h recall or FFQ data. Not meeting AI was defined as having inadequate intake in both the 24-h recall and FFQ or inadequate intake assessed by one tool and missing data for the other. Data was deemed missing if participants did not complete either of the dietary assessment tools.

All data manipulation, visual analyses, and statistical explanations were performed using STATA/IC, version 15.1 [35].

## 3. Results

### 3.1. Participant Characteristics

Of the 434 women recruited to the study, 204 were eligible for inclusion in this sub-set analysis. The baseline sociodemographic characteristics of the pregnant women from the Gomeroi Gaaynggal cohort included in this analysis are summarised in Table 1. The median maternal age was 24.1 years (IQR: 15.5–50.4). The majority of women had completed a high school education (Year 12) or less (53.2%), while approximately 18% were in some type of employment at the time of consent.

A summary of participants’ health characteristics and pregnancy outcomes are described in Table 2. A total of 18 women experienced a preterm birth, and 16 had HDP, while six women had both a preterm birth and HDP.

The majority of women reported a pre-pregnancy BMI over 25 kg/m^2^ (66.7%), with a median of 29.7 kg/m^2^ (IQR: 17.4–59.8). Participants who had a preterm delivery appeared to have a higher median BMI of 32.4 kg/m^2^ (IQR: 21.3–59.8), although data was only available for half of the group. In total, 29.0% of women reported having smoked at some point during their pregnancy, compared with 47.1% of those with preterm birth and 20.0% of those with HDP. Women who had either preterm birth or HDP had a higher prevalence of diabetes (type 1, type 2, or gestational) at 27.8% and 37.5%, respectively, compared with 14.8% of the total.

### 3.2. Maternal Dietary Intakes

A summary of total nutrient intake from foods and supplements, as recorded by 24-h recall during the early stages of pregnancy, is provided in Table 3. This data was collected at a median gestational age of 21.7 weeks (IQR: 5.3–37.6). The median daily energy intake of those included in this analysis was 7443 kJ (IQR: 710–16,992 kJ/day), with 48.3% of women meeting the recommended proportion of daily energy intake from total fat, known as the Acceptable Macronutrient Distribution Range (AMDR), with a median % energy from the fat of 34.5% (IQR: 6.8–65.3%). According to the 24-h recall analysis, only 55.2% and 28.7% of women met the AI for ALA and LA, respectively, while 37.1% of women met the AI for total EPA + DPA + DHA during pregnancy with a median intake of 84.4 mg/day (IQR: 0–588.3). By contrast, 50% of women with HDP meet the AI for total EPA + DPA + DHA with a median intake of 170.3 mg/day (IQR: 32.9–588.3). The data shows that none of the supplements taken during pregnancy by the women in this cohort contained any *n*-3 LC-PUFAs or other FAs (see Appendix A).

Nutrient intakes as assessed by FFQ, which did not assess supplementation during the third trimester of pregnancy, are summarised in Table 4, reporting intakes of the same key nutrients as in Table 3. The percentage of those women who met the AI for ALA and LA was similar to that recorded by the 24-h recall during the earlier stages of pregnancy at 59.8% and 32.0%, respectively. However, the total combined intake of EPA, DPA, and DHA, as assessed by FFQ, was higher, with a median intake of 186.7 mg/day (IQR: 20.6–643.9) and 81.4% of women meeting the AI.

### 3.3. Red Blood Cell Analysis

The FA profile of RBC membranes at each trimester of pregnancy is described in Table 5. Notably, 91.4% of total observations contained 0.0% ALA in the blood at any stage of gestation with a median of 0.0% (IQR: 0.0–0.6%). Saturated FAs account for the largest overall percentage of FAs in these blood samples at 43.2% (IQR: 41.1–71.9), with *n*-6 LC-PUFAs the next largest fraction at 27.9% (IQR: 4.7–32.9).

Figure 1 highlights the fraction of key LC-PUFAs in the maternal blood samples of women in the cohort who experienced preterm birth. Figure 2 highlights those who had HDP. The majority of pregnancies in the current sample were uncomplicated and indicated little change in the relative concentration of each FA in RBC membranes over the pregnancy. For those with preterm birth, the exception was for EPA and DHA, and therefore, total *n*-3 LC-PUFA concentrations. The trendlines appear to show an increase in the concentration of these *n*-3 LC-PUFAs across gestation in those with an uncomplicated pregnancy and a decrease in those who had a preterm birth. Those with HDP appear to have higher fractions of total *n*-3 LC-PUFAs and LA in their maternal RBC membranes than those who did not. Conversely, those with a preterm birth presented indications of lower concentrations of all the LC-PUFAs throughout the pregnancy compared with those who did not. The Omega-3 Index of women in the cohort is shown in Figure 3, highlighting those who had preterm birth and those who had HDP. The median Omega-3 Index for the whole cohort was 5.5% (IQR: 0.0–9.0). Appendix A illustrates the fractions of total *n*-3 LC-PUFAs, LA, and ALA compared with those meeting the AI for each, which shows no clear correlation between dietary intakes and FA concentrations in RBC membranes.

## 4. Discussion

This descriptive analysis examined the dietary intake of FAs and the FA profile of RBC membranes in the women of the Gomeroi Gaaynggal cohort during pregnancy. To the best of our knowledge, this is the first study to describe these factors in an Indigenous Australian population. The results highlight that most women from this cohort were meeting the national pregnancy NRV recommendations for intakes of total EPA, DPA, and DHA from the diet. While only over half of the women were meeting the guidelines for ALA, over 90% of the women included in this analysis had no detectable ALA in their RBC membranes at any stage throughout their pregnancy. Additionally, none of the supplements used by the participants contained any *n*-3 LC-PUFAs and only negligible levels of essential LC-PUFAs, ALA, and LA. The median Omega-3 Index for women in the cohort was 5.5%, with values ranging from 0.0–9.0%. When the LC-PUFA fractions from RBC membranes were compared for those with and without pregnancy complications, there appeared to be a decline in the concentration of both EPA and DHA throughout gestation in those women who had preterm birth. In contrast, there was no visible trend when evaluating the same LC-PUFA fractions in those who experienced HDP.

There is the limited literature reporting the dietary intakes of *n*-3 LC-PUFAs in pregnant women in Australia, and none specifically examining Indigenous women. An analysis of *n*-3 LC-PUFA intakes in pregnant women participating in the Australian Longitudinal Study on Women’s Health (ALSWH) reported a mean combined daily intake of EPA, DPA, and DHA of 336.2 mg/day (sd, 379.1) [12]. This is higher than the median intakes of women in the current Gomeroi Gaaynggal study, at 84.4 mg/day and 186.7 mg/day for intakes assessed by 24-h recall and FFQ, respectively. The reported mean ALA intake alone for pregnant women in the ALSWH, however, was equal to the median intake recorded by both the 24-h recall and FFQ in the Gomeroi Gaaynggal analysis at 1.1 mg/day [12]. It is worth noting that the data used in the ALSWH study was collected in 2003, and the population studied was considered to be a representative sample of Australian women aged 25–30 years at that time, with the total proportion of Indigenous women not reported. The sociodemographic and geographic differences between these two cohorts, along with the timing of data collection, could contribute to the inconsistencies in reported intakes. Moreover, while there has since been an increased awareness and emphasis on the role of *n*-3 LC-PUFAs in pregnancy [6], particularly among health researchers, public health recommendations simultaneously encourage pregnant women to exercise caution when eating fish [36,37].

The evidence describing *n*-3 LC-PUFA intakes in pregnant populations internationally is also limited and indicates varied intakes. A similar-sized study of pregnant women in Norway published in 2020 found that only 29.1% of participants were meeting Norwegian national recommendations to eat seafood at dinner two–three times per week [38]. This result aligns with the broader intake of Norwegian women aged 30–39 as reported in their national dietary survey of 2010–2011 [38]. However, the majority of pregnant women from this study (approximately 77%) were taking an *n*-3 LC-PUFA supplement at some point during their pregnancy, highlighting an understanding of the importance of *n*-3 LC-PUFAs for maternal health in this population [38]. In contrast, a small study in Belgium found that in 29 pregnant women, the median daily *n*-3 FA intake was 1.72 g, with an *n*-6/*n*-3 ratio of 8.78 [39]. The median intakes of EPA and DHA in these women were 120.0 mg/day and 150.0 mg/day, respectively, with 24.6% reportedly consuming a supplement [39]. A German study analysing the FA distribution of maternal and fetal blood found that fish was being consumed less than once a week, with only 20% of women using an *n*-3 LC-PUFA supplement during pregnancy [40]. The overall fat intake of this group was high at 45% of total energy intake; however, <1% (1.67 g/day) of this was from *n*-3 LC-PUFAs [40]. The dietary pattern of these women is described as an omnivore, with participants reporting that their dietary behaviours did not change while pregnant [40]. Interestingly, this study also found that the relative amount of *n*-3 LC-PUFAs in the maternal blood was no higher in those women who did use a supplement versus those who did not during pregnancy [40].

Most studies examining the relationship between dietary intake of FAs and corresponding levels in RBC membranes during pregnancy have been unable to draw a direct correlation between the two. The Norway study is the first to demonstrate any such relationship and used a principal component analysis to illustrate these findings [38]. Given the limited evidence to support an association between the two, it may be assumed that other factors also influence the FA profile of maternal blood. Additionally, despite over half of the women in the Gomeroi Gaaynggal cohort meeting the national recommendation for ALA intake from food, over 90% had no detectable ALA in their RBC membranes at any stage during their pregnancy. While previous studies analysing the FA profile of maternal blood have reported the overall percentage of ALA in their respective populations, they provide no further comment on the significance of these findings, focusing mainly on the roles of EPA and DHA. Of the other maternal studies located, all reported higher RBC membrane concentrations of ALA than what has been observed in the Gomeroi Gaaynggal cohort, with mean concentrations ranging from 0.10–0.29% [38,39,40,41,42,43]. One study in Japan reported a median ALA fraction of 0.2% with an IQR of 0.0–2.7%, confirming that concentrations of 0.0% are not isolated finding [43]. This is not altogether unsurprising, as ALA is a known precursor for longer chain PUFAs, is also utilised for β-oxidation, and is, therefore, not commonly stored [44,45]. However, the metabolism of ALA to EPA or DHA is considered to be poor [46]. This can be mediated by the amount consumed, with Goyens et al. [47] reporting that a low intake of LA increased ALA metabolism to EPA, while a high intake of ALA increased DHA production. Additionally, metabolism is improved in women compared to men, with oestrogen likely upregulating this metabolic pathway [45]. In fact, due to the high levels of DHA required for fetal development, metabolic adaptation to upregulate DHA production, including higher conversion of ALA, has been proposed during pregnancy as increased intake likely does not cover the additional requirements [45,48]. The low consumption of LA in this cohort, in combination with increased circulating oestrogen during pregnancy, may explain the low levels of ALA, as it was utilised as a precursor for LC-PUFA conversion. However, research into ALA metabolism in pregnancy and fetal development is limited and may play a larger role than a metabolic precursor [49].

There has been growing evidence linking higher RBC membrane concentrations of EPA and DHA to reduced risk of some pregnancy complications, including preterm birth and preeclampsia. The Omega-3 Index has been shown to be a reliable marker of EPA and DHA status, with an Omega-3 Index of 8–11% linked to a lower risk of cardiovascular disease [50]. The limited literature on Omega-3 Index suggests that it may be an appropriate marker during pregnancy for reduced risk of complications such as preterm birth [50]. The median Omega-3 Index of 5.5% recorded for the women in the Gomeroi Gaaynggal study analysis is not dissimilar to that of other pregnant population samples from studies in Belgium, Germany, and Norway [38,39,40]. With rates of preterm birth on the rise in many parts of the world, there have been calls to screen for the Omega-3 Index, either pre-conception or during the very early stages of pregnancy, to identify those at the highest risk [50]. However, given the weak link between dietary intake and FA concentrations identified in blood, further research is needed to better understand the factors resulting in lower levels in those individuals. Notably, a German study published in 2019, claiming to have produced the largest available database of FA analyses, stated that an Omega-3 Index of <2.0% is not possible in humans [51]. However, 21 of the women included in this analysis recorded an Omega-3 Index of <2.0%, with nine of the women at 0.0%. This is not the only time an Omega-3 Index of <2.0% has been reported in a study of pregnant women. Results from a 2020 study in Norway found that one participant had an Omega-3 Index of 1.93% [38]. These low values may be due to the process of analysing erythrocyte membrane fatty acids requiring measuring the peaks of the chromatograph. The software will only allow the measurement of peaks that meet a minimum requirement (area under the peak). It is possible that these participants had peaks for EPA and DHA that were too small to be detected. This may equally apply to ALA findings. Both findings would indicate that more research is required to better understand the implications of these levels and if there are any discrepancies in methodology between studies.

The results of the Gomeroi Gaaynggal study analysis highlight a trend towards lower concentrations of EPA and DHA in the later stages of pregnancy in those women who had preterm birth. By contrast, there is no obvious trend in the fractions of *n*-3 LC-PUFAs in women with HDP across gestation. Two high-quality systematic reviews found high-quality evidence that preterm birth rates are lower in those who consumed dietary and/or supplemental *n*-3 LC-PUFAs during their pregnancy compared with those who did not [6,52]. The 2018 Cochrane review, however, found only low-quality evidence, indicating a potential decrease in the rate of preeclampsia in those who consumed *n*-3 LC-PUFAs, whilst the more recent 2023 systematic review and meta-analysis found them to be protective [6,52]. While the reviews don’t compare EPA and DHA levels in the blood with these pregnancy complications, the findings are aligned with those observed in the Gomeroi Gaaynggal study analysis. It is also worth noting that for the six women included in this evaluation who had both a preterm birth and HDP, preterm birth may have been related to HDP. This possibility could skew these findings and likely result in a more obvious decline of EPA and DHA levels in those with preterm birth.

The limitations of the current study must be acknowledged. Dietary assessments are prone to measurement errors when relying on memory recall to estimate intakes. FFQs looking at intake over the previous 3–6 months can lead to an overrepresentation of recently consumed foods and a general overestimation of intake [53], which was addressed in this analysis by removing any entries with an implausible energy intake. The FFQ used in this study has also been shown to be a poor measure of fat and oil consumption, as the food list is limited and does not include specific kinds of margarine and vegetable oils [31]. Additionally, it doesn’t include *n*-3 LC-PUFA intakes from supplementation [31], which has only been captured in the current study using the 24-h recall data. Conversely, a single day of 24-h recall data may omit foods that are consumed less frequently, such as seafood, a common source of EPA and DHA, resulting in an underrepresentation of the usual intake of these nutrients. The gestational age at which each of the data sets used for this study (24-h recall, FFQ and maternal blood samples) were collected was not aligned and, therefore, cannot be triangulated. Additionally, a dietary analysis was unavailable for each blood sample and vice versa. All of these factors make it difficult to evaluate relationships between intake and the blood concentrations of *n*-3 LC-PUFAs associated with the pregnancy complications being assessed in this analysis. Due to the need to protect the anonymity of participants, it was not possible to report a specific diabetes diagnosis for each of the women (type 1, type 2, or gestational). This may have provided more insight into the results as each group has differing levels of clinical risk during pregnancy and may also have an influence on gestational duration. In addition, the sample size is relatively small, especially for those who had either a preterm birth and/or HDP. As all women participating in the analysis resided in a single regional or rural town of inland NSW, the findings may not be generalisable to broader Indigenous Australian populations. Despite the limitations of the dietary assessment tools used in this study, both the AES FFQ and the ASA24 are validated tools that have been deemed reliable measure of population intakes. To the best of our knowledge, this is the first study to analyse the dietary intake and FA profile of RBC membranes in a pregnant Indigenous population. The role of *n*-3 LC-PUFAs in maternal health is widely under-researched and even less so among Indigenous populations. This data will add to the gap of available knowledge in this area that, in time, may contribute to the development of clearer EAR and RDI guidelines for pregnant women. A better understanding of the role of *n*-3 LC-PUFAs in pregnancy and the factors associated with RBC membrane concentrations can lead to improved health outcomes for mother and baby and more equitable outcomes for Indigenous Australian populations.

### Practical Implications

Despite the limitations in being able to correlate intake of *n*-3 LC-PUFAs and RBC membrane concentrations, it is still advised that women from the Gomeroi Gaaynggal cohort aim to achieve the current Australian NRVs for these nutrients, as this is the best available evidence to support optimal health outcomes during pregnancy. The findings of the current study suggest that pregnant women from this population may benefit from increasing their intake of *n*-3 LC-PUFA-rich foods and/or choosing supplements to increase intake where appropriate. Following the results of the 2018 Cochrane review [6] and subsequent randomised control trials, the International Society for the Study of Fatty Acids and Lipids (ISSFAL) published a statement in September 2022 that advises women with a low baseline level of *n*-3 LC-PUFAs at the start of their pregnancy may benefit most from supplementation of approximately 1000 mg of total DHA + EPA to reduce the risk of preterm birth [54].

Food sources of EPA, DPA, and DHA that are accessible to women of the Gomeroi Gaaynggal cohort include canned tuna, mackerel, sardines, and, to a lesser extent, frozen fillets of Atlantic Salmon, which are more expensive and less available. Traditional Indigenous Australian foods, such as turtle and yabby, are also rich in *n*-3 LC-PUFAs [28]. The levels of mercury found in these marine foods is generally lower than some other sources and have been deemed safe for consumption during pregnancy [36]. Sources of ALA-rich foods may also be accessible and include canola oils and spreads, along with various nuts and seeds. Small amounts of these foods could help women to meet the NRV recommendation for ALA intake during pregnancy.

## 5. Conclusions

This study adds to the body of the literature regarding the role of *n*-3 LC-PUFAs during pregnancy, presenting data from a unique Indigenous cohort. While there is some evidence to support an association between higher intakes of *n*-3 LC-PUFA consumption during pregnancy and lower rates of preterm birth and preeclampsia, further research is needed to draw a definitive conclusion. Further research is needed regarding links between dietary intake, levels of FAs in the blood, metabolism of these FAs during pregnancy, and their relationship with pregnancy outcomes such as preterm birth and preeclampsia. Additionally, more research in other populations is needed to understand the independent role of ALA during fetal development and the significance of ALA levels found in the maternal blood of women in the Gomeroi Gaaynggal cohort. A better understanding of the role and acceptability of traditional Indigenous *n*-3 PUFA food sources during pregnancy may also assist in guiding culturally safe recommendations for this cohort.

## Figures and Tables

**Figure 1 nutrients-15-01943-f001:**
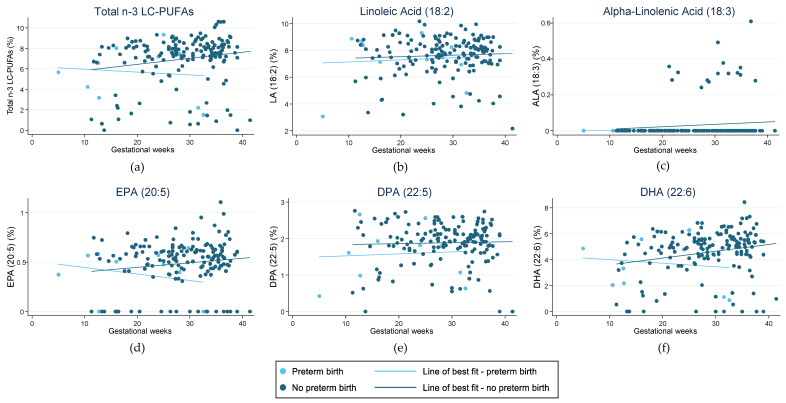
Fraction of LC-PUFAs (long-chain polyunsaturated fatty acids) in red blood cell membranes in pregnant women from the Gomeroi Gaaynggal cohort and those who experienced preterm birth: (**a**) Fraction of total *n*-3 LC-PUFAs (Omega-3 long-chain polyunsaturated fatty acids) (EPA, DPA and DHA); (**b**) Fraction of linoleic acid (18:2); (**c**) Fraction of alpha-linolenic acid (18:3); (**d**) Fraction of EPA (eicosapentaenoic acid) (20:5); (**e**) Fraction of DPA (docosapentaenoic acid) (22:5); (**f**) Fraction of DHA (docosahexaenoic acid) (22:6).

**Figure 2 nutrients-15-01943-f002:**
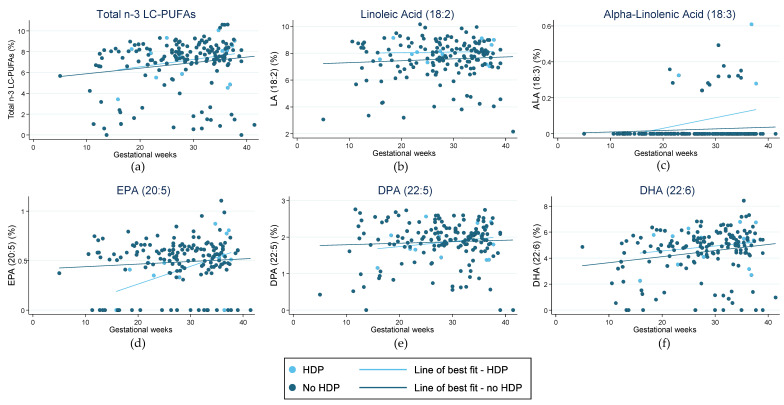
Fraction of LC-PUFAs (long-chain polyunsaturated fatty acids) in red blood cell membranes in pregnant women from the Gomeroi Gaaynggal cohort and those who experienced hypertension during pregnancy (HDP): (**a**) Fraction of total *n*-3 LC-PUFAs (Omega-3 long-chain polyunsaturated fatty acids) (EPA, DPA and DHA); (**b**) Fraction of linoleic acid (18:2); (**c**) Fraction of alpha-linolenic acid (18:3); (**d**) Fraction of EPA (eicosapentaenoic acid) (20:5); (**e**) Fraction of DPA (docosapentaenoic acid) (22:5); (**f**) Fraction of DHA (docosahexaenoic acid) (22:6).

**Figure 3 nutrients-15-01943-f003:**
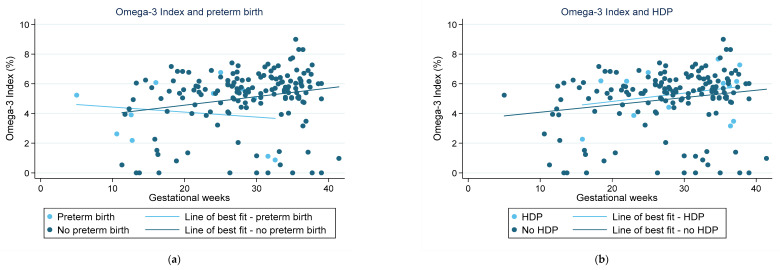
Omega-3 Index recorded from red blood cell membranes in pregnant women from the Gomeroi Gaaynggal cohort: (**a**) Those who experienced a preterm birth; (**b**) Those who experienced HDP (hypertension during pregnancy).

**Table 1 nutrients-15-01943-t001:** Sociodemographic characteristics of pregnant women in the Gomeroi Gaaynggal cohort (*n* = 204).

Characteristic	*n*	(%)
**Maternal Indigenous identity**		
Indigenous	167	(81.9%)
Carrying an Indigenous child	37	(18.1%)
**Level of education**		
<Year 10	18	(12.1%)
Year 10 or equivalent	60	(40.3%)
Year 12 or equivalent	31	(20.8%)
Trade/apprenticeship	23	(15.4%)
University degree	10	(6.7%)
Currently studying	7	(4.7%)
(*missing*)	55	
**Employment status**		
Full-time	16	(12.3%)
Part-time	10	(7.7%)
Casual	9	(6.9%)
No employment	90	(69.2%)
Other	5	(3.9%)
(*missing*)	74	
**Income**		
<$20,000	59	(54.1%)
$20,001–40,000	19	(17.4%)
$40,001–70,000	19	(17.4%)
>$70,001	12	(11.0%)
(*missing*)	95	

**Table 2 nutrients-15-01943-t002:** Comparison of health characteristics and pregnancy outcomes of Indigenous pregnant women in the Gomeroi Gaaynggal cohort with or without a preterm birth or hypertension during pregnancy.

	Uncomplicated (*n* = 176)		Preterm Birth * (*n* = 18)		HDP * (*n* = 16)		Total (*n* = 204)	
	*n* (%) or Median (IQR)		*n* (%) or Median (IQR)		*n* (%) or Median (IQR)		*n* (%) or Median (IQR)	
**Characteristic**								
Age	24.0	(15.5–50.4)	25.5	(18.2–34.4)	24.0	(16.7–40.8)	24.2	(15.5–50.4)
Pre-pregnancy BMI (kg/m^2^)	29.8	(17.4–52.0)	32.4	(21.3–59.8)	27.7	(22.1–43.5)	29.7	(17.4–59.8)
(*missing*)	69		9		6		81	
Parity								
0	19	(16.7%)	3	(25.0%)	2	(20.0%)	23	(17.3%)
1–2	68	(59.6%)	6	(50.0%)	5	(50.0%)	78	(58.6%)
3+	27	(23.7%)	3	(25.0%)	3	(30.0%)	32	(24.1%)
(*missing*)	62		6		6		71	
Smoking status								
Current	44	(28.0%)	8	(47.1%)	3	(20.0%)	53	(29.0%)
Non-smoker	113	(72.0%)	9	(52.9%)	12	(80.0%)	130	(71.0%)
(*missing*)	19		1		1		21	
Gestational age (weeks)	39.2	(37–43)	36.1	(30.3–36.5)	37.8	(30.3–41.4)	39.0	(30.3–43.0)
(*missing*)	22		-		-		22	
Birth weight (g)	3455	(1719–5430)	2612	(910–3550)	2915	(1150–4125)	3372.5	(910–5430)
(*missing*)	22		-		-		22	
Diabetes status								
Nil	136	(87.7%)	13	(72.2%)	10	(62.5%)	156	(85.2%)
T1DM, T2DM or GDM	19	(12.3%)	5	(27.8%)	6	(37.5%)	27	(14.8%)
(*missing*)	21		-		-		21	

BMI: Body Mass Index. GDM: Gestational diabetes mellitus. HDP: Hypertension during pregnancy. IQR: Interquartile range. T1DM: Type 1 diabetes mellitus. T2DM: Type 2 diabetes mellitus. Uncomplicated: Those without preterm birth or HDP. * Six women had both a preterm birth and HDP.

**Table 3 nutrients-15-01943-t003:** Comparison of nutrient intake from foods and supplements ^ of pregnant women in the Gomeroi Gaaynggal cohort with or without a preterm birth or hypertension during pregnancy, as recorded by 24-h recall.

24-h Recall Dietary Analysis													
		Uncomplicated (*n* = 125)			Preterm Birth * (*n* = 13)			HDP * (*n* = 10)			Total (*n* = 143)		
		Median	(IQR)	Meeting NRV	Median	(IQR)	Meeting NRV	Median	(IQR)	Meeting NRV	Median	(IQR)	Meeting NRV
**Nutrient**	**NRV**												
Energy (kJ/day)	5200–15,600 (EER)	7443	(710–16,992)	-	6959	(1939–11,705)	-	7227	(5754–14,095)	-	7443	(710–16,992)	-
Energy total Fat (%)	20–35% (AMDR)	34.4	(6.8–57.8)	48.0%	34.7	(19.4–65.3)	53.8%	34.8	(19.3–54.0)	50.0%	34.5	(6.8–65.3)	48.3%
Energy total SFAs (%)	8–10% (AMDR)	13.9	(1.2–31.0)	9.6%	14.3	(8.4–25.3)	15.4%	15.3	(5.7–25.3)	0.0%	14.0	(1.2–31.0)	8.5%
Total *n*-3 LC-PUFAs (mg/day)	115 (AI)	87.0	(0–588.7)	37.6%	82.8	(4.7–588.3)	38.5%	170.3	(32.9–588.3)	50.0%	84.4	(0–588.7)	37.1%
EPA	-	12.9	(0–172.4)	-	18.1	(0–186.1)	-	48.0	(4.7–130.5)	-	13.1	(0–186.1)	-
DPA	-	44.4	(0–212.0)	-	45.7	(4.7–174.1)	-	46.8	(16.1–174.1)	-	44.6	(0–212.0)	-
DHA	-	16.2	(0–343.7)	-	11.6	(0–415.4)	-	23.0	(0–415.4)	-	15.5	(0–415.4)	-
ALA (18:3) (g/day)	1.0 (AI)	1.1	(0.1–4.6)	56.8%	0.9	(0.1–2.6)	38.5%	1.0	(0.5–1.9)	50.0%	1.1	(0.1–4.6)	55.2%
LA (18:2) (g/day)	10 (AI)	7.6	(0.3–20.9)	29.6%	5.5	(1.5–25.1)	15.4%	5.7	(3.5–13.6)	20.0%	7.6	(0.3–25.1)	28.7%
SFAs (g/day)	-	28.5	(2.8–102.5)	-	25.7	(5.0–52.9)	-	32.1	(8.9–64.9)	-	28.5	(2.8–102.5)	-
MUFAs (g/day)	-	27.2	(2.2–81.6)	-	25.4	(2.9–61.7)	-	25.1	(11.2–55.2)	-	26.7	(2.2–81.6)	-
PUFAs (g/day)	-	9.4	(0.4–25.7)	-	6.4	(1.7–28.1)	-	7.1	(4.6–15.1)	-	9.4	(0.4–28.1)	-
*n*-6/*n*-3 ratio	-	5.9	(2.1–16.6)	-	6.1	(3.0–10.4)	-	5.4	(3.4–10.5)	-	5.9	(2.1–16.6)	-

AI: Adequate Intake. ALA: Alpha-linolenic acid. AMDR: Acceptable Macronutrient Distribution Range. DHA: Docosahexaenoic acid. DPA: Docosapentaenoic acid. EER: Estimated Energy Requirement. EPA: Eicosapentaenoic acid. HDP: Hypertension during pregnancy. IQR: Interquartile range. LA: Linoleic acid. MUFAs: Monounsaturated fatty acids. NRV: Nutrient Reference Value. *n*-3 LC-PUFAs: Omega-3 long-chain polyunsaturated fatty acids. PUFA: Polyunsaturated fatty acids. SFAs: Saturated fatty acids. Uncomplicated: Those without preterm birth or HDP. ^ Supplemental nutrient intake in isolation has been tabulated separately in the Appendix A. * 5 women had both a preterm birth and HDP.

**Table 4 nutrients-15-01943-t004:** Comparison of nutrient intake from foods of Indigenous pregnant women in the Gomeroi Gaaynggal cohort with or without a preterm birth or hypertension during pregnancy, as recorded by food frequency questionnaire during the third trimester.

FFQ Dietary Analysis													
		Uncomplicated (*n* = 86)			Preterm Birth (*n* = 4)			HDP * (*n* = 7)			Total (*n* = 95)		
		Median	(IQR)	Meeting NRV (%)	Median	(IQR)	Meeting NRV (%)	Median	(IQR)	Meeting NRV (%)	Median	(IQR)	Meeting NRV (%)
**Nutrient**	**NRV**												
Energy (kJ/day)	5200–15,600 (EER)	8632.5	(5031–19,809)	-	9416	(6918–13,940)	-	10,482	(4739–14,401)	-	8689	(4739–19,809)	-
Energy total Fat (%)	20–35% (AMDR)	34.8	(24.3–42.7)	53.5%	41.5	(33.9–45.1)	25.0%	35.5	(23.1–43.7)	28.6%	34.9	(23.1–45.1)	50.5%
Energy total SFAs (%)	8–10% (AMDR)	14.1	(8.2–18.9)	4.7%	17.9	(12.4–19.4)	0.0%	15.0	(8.7–17.1)	14.3%	14.1	(8.2–19.4)	5.3%
Total *n*-3 LC-PUFAs (mg/day)	115 (AI)	188.8	(54.6–643.9)	82.6%	132.6	(76.6–536.9)	50.0%	238.1	(20.6–536.9)	71.4%	186.7	(20.6–643.9)	81.1%
EPA	-	49.5	(8.0–194.3)	-	36.2	(23.5–139.1)	-	56.0	(3.2–139.1)	-	49.4	(3.2–194.3)	-
DPA	-	84.1	(21.0–255.8)	-	79.9	(40.7–267.4)	-	91.2	(11.5–267.4)	-	84.6	(11.5–267.4)	-
DHA	-	49.1	(4.0–290.4)	-	16.6	(12.3–130.4)	-	65.0	(5.8–130.4)	-	48.6	(4.0–290.4)	-
ALA (18:3) (g/day)	1.0 (AI)	1.1	(0.5–2.2)	59.3%	1.3	(0.8–2.7)	50.0%	1.2	(0.4–2.7)	57.1%	1.1	(0.5–2.7)	59.0%
LA (18:2) (g/day)	10 (AI)	8.0	(3.7–19.2)	32.6%	10.9	(6.3–16.5)	50.0%	9.5	(4.3–16.5)	28.6%	8.1	(3.7–19.2)	32.6%
SFAs (g/day)	-	33.1	(11.9–81.3)	-	48.6	(23.2–64.5)	-	41.8	(11.1–64.5)	-	35.2	(11.1–81.3)	-
MUFAs (g/day)	-	33.1	(14.3–90.1)	-	40.8	(25.8–59.9)	-	39.1	(10.5–64.3)	-	33.2	(10.5–90.1)	-
PUFAs (g/day)	-	9.7	(4.5–21.8)	-	12.7	(7.4–20.7)	-	11.5	(4.9–20.7)	-	9.8	(4.5–21.8)	-
*n*-6/*n*-3 ratio	-	6.3	(3.9–9.1)	-	6.8	(5.2–8.4)	-	6.7	(5.2–9.3)	-	6.3	(3.9–9.3)	-

AI: Adequate Intake. ALA: Alpha-linolenic acid. AMDR: Acceptable Macronutrient Distribution Range. DHA: Docosahexaenoic acid. DPA: Docosapentaenoic acid. EER: Estimated Energy Requirement. EPA: Eicosapentaenoic acid. FFQ: Food Frequency Questionnaire. HDP: Hypertension during pregnancy. IQR: Interquartile range. LA: Linoleic acid. MUFAs: Monounsaturated fatty acids. NRV: Nutrient Reference Value. *n*-3 LC-PUFAs: Omega-3 long-chain fatty acids. PUFAs: Polyunsaturated fatty acids. SFAs: Saturated fatty acids. Uncomplicated: Those without a preterm birth or HDP. * 2 women had both a preterm birth and HDP.

**Table 5 nutrients-15-01943-t005:** Fraction of long-chain polyunsaturated fatty acids and full fatty acid profile of red blood cell membranes at each trimester of pregnancy in women from the Gomeroi Gaaynggal cohort.

	Trimester						Total (*n* = 174)	
	One (1–12 Wks) (*n* = 4)		Two (13–28 Wks) (*n* = 85)		Three (29 Wks–Birth) (*n* = 85)			
	Median	(IQR)	Median	(IQR)	Median	(IQR)	Median	(IQR)
**Nutrient**								
LA (18:2) (%)	7.2	(3.1–8.9)	7.7	(3.2–10.2)	8.0	(2.2–10.0)	7.9	(2.2–10.2)
ALA (18:3) (%)	0.0	(0.0)	0.0	(0.0–0.4)	0.0	(0.0–0.6)	0.0	(0.0–0.6)
EPA (20:5) (%)	0.5	(0.0–0.8)	0.5	(0.0–1.0)	0.6	(0.0–1.1)	0.5	(0.0–1.1)
DPA (22:5) (%)	1.1	(0.4–2.8)	2.0	(0.0–2.7)	2.0	(0.0–2.7)	2.0	(0.0–2.8)
DHA (22:6) (%)	2.6	(0.5–4.9)	4.9	(0.0–8.1)	5.1	(0.0–8.4)	5.0	(0.0–8.4)
SFAs (%)	48.9	(43.9–60.9)	43.1	(41.1–71.9)	43.3	(41.1–70.0)	43.2	(41.1–71.9)
MUFAs (%)	20.2	(17.6–28.4)	20.9	(18.4–28.1)	20.9	(8.2–30.7)	20.9	(8.2–30.7)
*n*-6 LC-PUFA (%)	25.9	(9.7–31.8)	28.2	(4.7–31.5)	27.7	(5.3–32.9)	27.9	(4.7–32.9)
*n*-3 LC-PUFA (%)	5.0	(1.1–6.7)	7.6	(0.0–10.8)	7.7	(0.0–10.6)	7.6	(0.0–10.8)
*n*-6/*n*-3 ratio	5.3	(4.7–9.2)	3.7	(0.0–15.2)	3.6	(0.0–13.4)	3.7	(0.0–15.2)
Omega-3 Index	3.3	(0.5–5.2)	5.5	(0.0–8.7)	5.7	(0.0–9.0)	5.5	(0.0–9.0)

ALA: Alpha-linolenic acid. DHA: Docosahexaenoic acid. DPA: Docosapentaenoic acid. EPA: Eicosapentaenoic acid. FA: Fatty acid. IQR: Interquartile range. LA: Linoleic acid. LC-PUFA: Long-chain polyunsaturated fatty acid. MUFAs: Monounsaturated fatty acids. SFAs: Saturated fatty acids. Wks: Weeks.

## Data Availability

The data reported in this study are available on request from the corresponding author and the Gomeroi Gaaynggal Advisory Committee. The data are not publicly available for ethical reasons.

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
