# Peer review of "Omega-3 Fatty Acids during Pregnancy in Indigenous Australian Women of the Gomeroi Gaaynggal Cohort"

_nutrients, 2023, doi:10.3390/nu15081943_

Round 1
Reviewer 1 Report
The Author report on the Omega-3 Fatty acids during pregnancy in Australian women of the Gomeroi cohort. The paper is well written and address an important topic during pregnancy.
1. In the section of pregnancy outcomes regarding the method used for HDP needs reviewing and reference of the method used as for example the definition of preeclampsia. Also Diastolic of 300 mg (/ 90mmgh) This section needs corrections
2. There is fundamental problem in the method used for food recall before and after 2018 this is a limitation of the study and has an impact on the validity of the results, making it very difficult to draw any conclusion. Although the Author report them separately but this has potential influence on the results validity.
3. The section on the statistical analysis is very complicated as they used both 24hrs recall and FFQ data in different sections, I would recommend getting biostatistician review on the method used the address this and they way they decide to present their data
4. The numbers of participants are generally small for preterm birth and HDP and despite this there is about 50% missing data so only reporting on 9 subjects in both categories. Although this has been addressed, the applicability of the data se can be questioned.
5.The maternal dietary intakes data was collected at median age of 21.7 week (IQR: 5.3-37.6) this will have an impact on the results and should be addressed in the discussion as it may contribute to the results findings. Also, the fact that FFQ didn’t assess supplementation.
6. The plausible causes of having no detectable ALA in the RBC membranes at any stage of the pregnancy despite adherence to ALA intake from food recommendation were not addressed extensively which will leave the reader puzzled about this finding, could it be related the method used in the food recall or data collections?
7. Some of the above valid points were addressed in the discussion section but needs to be more emphasized.
8. The conclusion needs to be rewritten as it was not specific conclusion of the study and there was a jump to recommendation before a solid study conclusion, this should also be reflected and changed in the in the abstract.
9. Suggest to add to recommendation of future studies with details about the ideal study to address the hypothesis used this study and what is learned from this study try to overcome it which will better inform the literature and the researcher.

Reviewer 2 Report
The authors present an article of a descriptive study that aimed to describe the dietary intake of LC-PUFAs in a cohort of Indigenous Australian women during their pregnancy. The authors use the Gomeroi Gaaynggal cohort. Dietary intake was measured through a 24-h food recall and food frequency questionnaire during the second and third trimester of pregnancy. A large percentage of women met the recommendation for EPA, DPA, and DHA only with diet, according to FFQ. However, most of them had no detectable ALA in RBC. There is no association between omega-3 consumption, preterm birth rates, and hypertension during pregnancy.
Abstract:
Line 28: “Most women in this cohort met national n-3 LC-PUFA recommendations” (according to FFQ), with 55% meeting alpha-linolenic acid (ALA) recommendations (according to 24h-recall)”. These results are inaccurate because two different tools measure the same results. Why did you decide to use FFQ? Both values should be considered.
Introduction:
The authors could explain the importance of determining the intake of LC-PUFA in this population and how different it can be from the rest of the Australian population (cooking culture, food access, economic status, education, etc.).
Materials and Methods:
The paper provides new information about omega-3 intake in a specific population of Australia. However, there is no statistical analysis of the results. I strongly recommend performing statistical analyses on the health characteristics of participants and incorporating the p-value in the table. Same for all results.
Discussion:
A complete explanation of why the omega-3 index was 0.0% in 9 women is required, as the authors did with RBC levels.
The authors registered different results of media intake between 24-h recall and FFQ. A complete explanation or hypothesis could help to understand the difference.
Line 560-562: the authors described a trend towards lower concentrations of EPA and DHA in preterm. Statistical analysis is necessary to determine if the difference among the groups impacts preterm birth.
The authors are aware of the study’s limitations which are mentioned in the text.
References:
Only 11 references are from 2018 onwards, representing 21% of recent publications.
Reviewer 3 Report
the authors propose an interesting report on the uses of an Australian population.
There are some points to clarify and improve:
- A table with the average values of kcal and micro and macronutrients should be reported and also an example of a typical day, eating habits can also be very different when compared to other continents
- It should be emphasized in the possible weaknesses that the method, although used, is not said to be connected to the quantity of DHA and EPA in the body.
- An opinion from the authors on the appropriateness of supplementing DHA and EPA would be useful, even if the study did not show a clear correlation with the supposed problems
Reviewer 4 Report
This is a very interesting and original article on omega-3 fatty acids during pregnancy in Indigenous Australian women.
The authors introduced the importance of omega-3 fatty acids during pregnancy including lower rates of preterm birth, preeclampsia and perinatal depression. They also provided recommendations on polyunsaturated fatty acids for pregnant women in Australia (i.e., AI target is 10 g/day for LA, 1.0 g/day for ALA and 115 mg/day for total EPA, DPA and DHA consumption). According to the 2012-13 Health Survey results, Indigenous Australian women of childbearing age met the AI for n-3 LC-PUFA intake during pregnancy with an average daily ALA intake of 1.3 g, and 204 mg of total EPA, DPA and DHA.
The methodology section is divided into several sections. The Study design and Dietary and Supplemental Intake of Omega-3 Fatty Acids are described in detail.
“diastolic pressure ≥ 300mmHg” in line 180 may be a typo. Check this!?
Results are clearly presented in tables and figures and correspond with the main text.
The discussion is comprehensive and well-written.
The paper complies with the field of the journal.
Reviewer 5 Report
The manuscript submitted for review deals with the interesting and important problem of omega-3 intake during pregnancy. The beneficial role of omega-3 fatty acids in the functioning of the body at different stages of development has long been the subject of much research, although further studies aimed at a deeper understanding of the mechanisms of this action conducted on different target groups are justified and needed.
I have some minor comments which are as follows:
line 65 - the authors stated that ALA is predominant among omega-3 in Western diet - I am not convinced this is true, and the reference provided seems not to support this claim
line 119 - two locations - one rural and one remote?
line 136 - regional and remote town, this seems as it was one town only (which however is explained later on), why terme "regional" and "remote" are used? Can't it be simply town 1 and town 2?
lines 190 -191 - data on diet collected between 2014-2019, what about data (participants) from previous years starting from 2009?
Table 3, line 355 "... nutrient intake from foods and supplements" - it is indicated below the table that supplemental nutrient intake is presented in supplemental materials, while it is not
Fig. 1 and Fig. 2 - captions - "....pregnant women (...) and those who experoenced ...." - were these 2 groups, pregnant women in total and those with preterm birth (Fig. 1) and HDP (Fig. 2)?
Round 2
Reviewer 2 Report
The authors addressed the comments satisfactorily.